# Digital Twin Model of Investment Cash Flows in Distributed Ledger Environment with Neural Network Forecasting

**Glushankov Kirill**[*],        **Sergey Evgenievich Barykin**[†],        **Daria Dinets**[‡].

## Abstract

The article examines the problem of formalizing investment cash flow in a distributed ledger environment. Within the framework of the digital transformation of financial relations, the cash flow of an investment project can be represented as a digital twin, recorded in the distributed ledger infrastructure and implemented through smart contracts. The aim of the study is to develop a mathematical model of the digital twin of investment cash flow and an algorithm for its forecasting using neural networks. Theoretical approaches to the interpretation of digital twins are systematized, and the limitations of the classical discounted cash flow model in relation to the digital environment are analyzed. A formalized model of digital cash flow is proposed, taking into account transaction fees of the distributed ledger, algorithmically accrued income, and an extended discount rate structure including technological and regulatory risk premiums. An algorithm for neural network forecasting of the digital twin is developed based on a feature vector integrating financial and infrastructure parameters. A comparative analysis of the digital and classical models is performed, which allowed establishing the structural modification of the investment process in the digital environment. The obtained results can be used in the valuation of digital financial assets and the construction of adaptive systems for forecasting their cash flows.

## 1   Introduction

The development of distributed ledger infrastructure and the digital financial assets market has led to the formation of new forms of representing investment processes in the digital environment. In the context of the digital transformation of economic relations, cash flows accompanying investments in production facilities acquire an additional functional form – a digital record in the distributed ledger. As a result, a digital twin of the investment cash flow is formed, possessing specific characteristics of circulation, accounting, and forecasting.

In classical financial theory, the cash flow of an investment project is considered as a set of cash inflows and outflows, discounted taking into account the time value of money and risk. At the same time, the legal structure of investment implies the transfer of ownership of the asset to the investor or the emergence of a debt claim. In the context of digital financial assets circulation, the investment process can be implemented without the transfer of ownership of the tangible object, and the investor's rights are recorded in the distributed ledger through smart contract algorithms.

A contradiction arises, determining a scientific gap: the digital twin of cash flow, operating in a distributed ledger, formally reproduces the parameters of the classical discounted cash flow model, however, it includes additional infrastructure and algorithmic components associated with the fee mechanisms of the distributed ledger (blockchain systems), decentralization of data storage, automation of settlements, and the specifics of determining the discount rate. Insufficient development

---

[*]Peoples' Friendship University of Russia named after Patrice Lumumba, Moscow, Russia. Email: kglush@rambler.ru

[†]Doctor of Economics, Professor, Peoples' Friendship University of Russia, Moscow, Russia. Email: sbe@list.ru.

[‡]Doctor of Economics, Associate Professor, Peoples' Friendship University of Russia named after Patrice Lumumba, Moscow, Russia. Email: dinets-da@rudn.ru.

of the theoretical model of the digital twin of investment cash flow hinders the correct assessment of its efficiency, risks, and predictive parameters. The scientific literature presents studies on digital twins, neural networks, and blockchain infrastructure, but a formalized model of the digital twin of an investment project's cash flow, considering the features of the distributed ledger and artificial intelligence tools, is absent.

The object of the research is the cash flows of investment projects in the digital environment of a distributed ledger.

The subject of the research is the economic and mathematical models of the digital twin of investment cash flow and algorithms for its forecasting using neural networks.

The aim of the study is to formalize the model of the digital twin of an investment project's cash flow in a distributed ledger and identify its distinctive characteristics compared to the classical discounted cash flow model.

To achieve this goal, the following tasks are solved:

- systematize theoretical approaches to the interpretation of digital twins in economics;
- formalize the model of the digital twin of investment cash flow;
- determine the structural differences between the digital model and the classical discounted cash flow model;
- develop an algorithm for forecasting the digital twin using neural networks;
- perform a comparative analysis of the digital and traditional models of investment cash flow.

The scientific novelty of the research lies in the development of a formalized model of the digital twin of investment cash flow, taking into account the infrastructure parameters of the distributed ledger, algorithmic features of smart contracts, and the application of neural networks for forecasting.

The practical significance of the work consists in the possibility of applying the developed model to assess the efficiency of digital financial assets backed by investments in production facilities of IT infrastructure.

## 2  LITERATURE REVIEW

### 2.1  THE CONCEPT OF DIGITAL TWIN IN ECONOMIC SYSTEMS

The concept of a digital twin was initially formed in engineering and production systems as a tool for virtual reproduction of a physical object for the purpose of monitoring, modeling, and optimizing its functioning. Subsequently, the concept was developed in economic research, where a digital twin is considered as a digital model reflecting the state and dynamics of economic processes.

In the work of Abramov V.I., Gordeev V.V., Stolyarov A.D. (2024), a digital twin is defined as a structured digital model that ensures the reproduction of an object's characteristics and its behavior over time. The authors highlight the architectural elements of a digital twin, including data collection, processing, and the formation of analytical conclusions. However, in this study, the digital twin is mainly interpreted as a model of a physical or production object, while the financial component is disclosed to a limited extent.

In the context of digital transformation of financial markets, there is a need to extend the interpretation of digital twins to the cash flows of investment projects. Cash flow in this context can be considered as a dynamic digital object, reproduced in a distributed ledger and subject to algorithmic processing.

### 2.2  CLASSICAL MODELS OF INVESTMENT PROJECT CASH FLOWS

The foundation of investment process analysis is the discounted cash flow theory. In the classical model, the cash flow of an investment project is defined as a sequence of cash inflows and outflows, discounted at a rate $r$, considering the time value of money and risk.

The methodological basis for evaluating investment projects is formed within the framework of capital budgeting theory and is represented by indicators such as net present value, internal rate of return, and payback period. This model assumes centralized transaction accounting, the availability of reliable historical data, and the possibility of determining the discount rate based on market parameters.

However, the classical model does not account for the specifics of digital financial assets circulation, the fee mechanisms of blockchain systems, algorithmic execution of obligations, and the infrastructure costs of distributed ledgers. This creates a theoretical gap between the traditional model of investment cash flow and its digital representation.

## 2.3 Blockchain Infrastructure and Digital Financial Assets

The development of distributed ledgers has led to the formation of a market for digital financial assets, whose rights are recorded as entries in the blockchain. Research by Tur'yan K.V. (2023) demonstrates that smart contracts ensure automation of settlements and reduce transaction costs characteristic of traditional intermediary models.

In the work of Carlos Núñez-Gómez et al. (2025), decentralization is considered as a factor in the redistribution of control and data storage functions between network nodes. This affects the structure of fee costs, risk distribution, and the algorithmics of transaction confirmation.

Janne Ruponen et al. (2025), developing the concept of cogitome introduced by Anokhin, propose the concept of a decentralized cogitome as a mechanism for processing and synchronizing financial information in a blockchain environment. Furthermore, the authors propose the concept of a logistikon as a tool for modeling digital processes and transforming physical economic operations into digital form.

Despite the significance of these studies, they lack a formalized model of investment cash flow as a digital twin integrated into a distributed ledger.

## 2.4 Application of Neural Networks in Financial Flow Forecasting

In works devoted to artificial intelligence in economics, neural networks are considered as a tool for identifying patterns in large volumes of financial data. Matyush I.V. (2024) analyzes the architecture of neural networks in relation to processing accounting information, including correspondences of cash accounts.

Modern machine learning models allow for considering the non-linearity of cash flow dynamics, adaptively adjusting forecasts as new data arrives, and forming scenario assessments. However, the application of neural networks to digital twins of investment cash flows in a distributed ledger remains an insufficiently researched area.

## 2.5 Identified Research Gap

The analysis of scientific literature allows for the following conclusions:

1. The concept of a digital twin is developed mainly for physical and production objects.

2. The classical theory of discounted cash flows does not account for the infrastructure features of a distributed ledger.

3. Blockchain technologies have been studied from the perspective of legal and technological architecture, but not in the context of formalizing investment cash flow as a digital twin.

4. Neural networks are used for forecasting financial indicators, but their integration with the digital twin model of investment cash flow is not theoretically systematized.

Thus, the formation of a mathematical model of the digital twin of investment cash flow in a distributed ledger and the development of an algorithm for its forecasting using neural networks represents an independent scientific task.

## 3  METHODOLOGY OF SCIENTIFIC RESEARCH

The methodological basis of the research is the synthesis of the provisions of the discounted cash flow theory, institutional analysis of digital financial assets, as well as methods of mathematical modeling and machine learning.

The work uses a systematic approach, which involves considering the investment cash flow as a dynamic economic system that functions over time and is transformed into digital form in a distributed ledger environment.

The research methodology includes the following interrelated stages:

1. Formalization of the classical model of investment cash flow based on discounting theory.
2. Identification of infrastructure parameters of the distributed ledger that affect the structure of cash flow.
3. Construction of a modified model of the digital twin of investment cash flow.
4. Development of an algorithm for neural network forecasting of the digital twin.
5. Comparative analysis of the classical and digital models.

The tools used include:

- methods of financial mathematics;
- elements of probability theory in interpreting risk parameters;
- functional modeling;
- the concept of a multilayer neural network;
- scenario analysis.

The key methodological assumption of the study is the following proposition: the digital twin of investment cash flow is not a simple replication of the classical model, but represents its structurally modified form, into which algorithmic, infrastructure, and fee parameters of the distributed ledger are integrated.

Further presentation is based on the principle of successive complication of the model – from the basic classical structure to the digital modification.

## 4  CLASSICAL MODEL OF INVESTMENT CASH FLOW

Consider an investment project with a planning horizon $T$.

Denote: $CF_t$ – net cash flow in period $t$; $I_0$ – initial investment costs; $r$ – discount rate; $T$ – calculation horizon.

The classical discounted cash flow model takes the form:

$$NPV = -I_0 + \sum_{t=1}^{T} \frac{CF_t}{(1+r)^t},$$

where $NPV$ is the net present value of the project.

Structurally, cash flow $CF_t$ is formed as the difference between inflows and outflows:

$$CF_t = Inflow_t - Outflow_t.$$

For investments in production facilities of IT infrastructure, cash flow includes:

- income from equipment rental;
- operating expenses;
- maintenance costs;

- tax payments;
- changes in working capital.

The classical model assumes:

1. centralized transaction accounting;
2. existence of ownership rights to the asset;
3. possibility of determining the rate $r$ based on market data;
4. absence of infrastructure fees of the distributed ledger.

This model is the baseline point of comparison for constructing the digital twin of investment cash flow.

# 5 FORMALIZED MODEL OF THE DIGITAL TWIN OF INVESTMENT CASH FLOW

## 5.1 GENERAL FORMULATION

Let an investment project be implemented through the issuance of digital financial assets (DFAs) backed by production facilities of IT infrastructure. Ownership of the object remains with the issuer, and the investor acquires the right to receive cash proceeds recorded in the distributed ledger.

Unlike the classical model, the digital twin of cash flow operates in a distributed ledger environment and includes infrastructure parameters of the blockchain system.

Denote $\widetilde{CF}_t$ as the digital twin of cash flow in period $t$; $\widetilde{NPV}$ as the present value of the digital twin; $r_d$ as the discount rate of the digital cash flow; $T$ – forecasting horizon. Then the present value of the digital twin takes the form:

$$\widetilde{NPV} = \sum_{t=1}^{T} \frac{\widetilde{CF}_t}{(1 + r_d)^t}.$$

## 5.2 STRUCTURE OF THE DIGITAL TWIN OF CASH FLOW

Unlike the classical model, the digital twin includes additional components due to the functioning of the distributed ledger.

Let us present the structure of the investor's digital cash flow:

$$\widetilde{CF}_t = P_t + I_t - TR_t - CR_t - EC_t,$$

where: $P_t$ – payments from leasing server equipment, accrued to the investor; $I_t$ – algorithmically calculated interest income; $TR_t$ – fees for transactions in the blockchain and conversion into national currency; $CR_t$ – fees for storage and maintenance of the digital asset; $EC_t$ – share of operating expenses for equipment maintenance.

## 5.3 FEATURES OF MODEL COMPONENTS

**1. Payments $P_t$.** Payments are formed based on the issuer's rental income. Their accrual is recorded by a smart contract and reflected in the distributed ledger. Unlike the classical model, the fulfillment of obligations is algorithmic in nature.

**2. Interest Income $I_t$.** Interest income depends on the issuer's financial results and is calculated automatically based on the terms of the DFA issuance. Thus, the investment cash flow combines features of investment and financial income.

**3. Transaction Fees $TR_t$.** Fees are determined by the parameters of the blockchain network and depend on:

- transaction size;
- network congestion;
- transaction confirmation algorithm.

$TR_t$ is a variable value and can be modeled as a function of transaction volume:

$$TR_t = f(V_t, \lambda_t),$$

where $V_t$ – transfer volume, $\lambda_t$ – the network congestion parameter.

**4. Storage Fees** $CR_t$**.** $CR_t$ reflect the costs of maintaining the distributed ledger infrastructure and storing the digital asset.

**5. Operating Expenses** $EC_t$**.** $EC_t$ represent the share of expenses for technical maintenance of server equipment, distributed between the issuer and the investor.

### 5.4 MODIFIED DISCOUNT RATE

The rate $r_d$ is formed considering additional risks of the digital environment:

$$r_d = r_f + \pi_m + \pi_{reg} + \pi_{tech},$$

where: $r_f$ – risk-free rate; $\pi_m$ – premium for market volatility of DFA; $\pi_{reg}$ – premium for regulatory risk; $\pi_{tech}$ – premium for technological risk.

In the absence of historical data, the rate can be determined based on the equipment's payback period, expressed as an annual percentage.

### 5.5 FUNDAMENTAL DIFFERENCES FROM THE CLASSICAL MODEL

The digital twin of investment cash flow differs from the classical model by:

1. absence of transfer of ownership of the tangible asset;
2. algorithmic fixation of obligations;
3. presence of infrastructure fees;
4. decentralized data storage;
5. extended structure of the discount rate.

Thus, the digital twin is a modified form of investment cash flow, integrated into a distributed ledger.

## 6 ALGORITHM FOR NEURAL NETWORK FORECASTING OF THE DIGITAL TWIN OF INVESTMENT CASH FLOW

### 6.1 GENERAL FORMULATION OF THE FORECASTING PROBLEM

The digital twin of investment cash flow in a distributed ledger is a time series of values $\widetilde{CF}_t$, formed based on blockchain transaction data, financial parameters of the issuer, and infrastructure indicators of the network.

The forecasting task is formulated as constructing a mapping:

$$\Phi : X_{t-k:t} \to \widehat{\widetilde{CF}}_{t+1:t+h},$$

where: $X_{t-k:t}$ – a set of observations and features over a window of length $k$; $\widehat{\widetilde{CF}}_{t+1:t+h}$ – forecast of the digital cash flow for horizon $h$; $\Phi$ – forecasting function, approximated by a neural network.

Unlike classical cash flow forecasting tasks, the input data is formed from decentralized sources and contains parameters reflecting the state of the blockchain infrastructure.

## 6.2 FORMATION OF THE INPUT FEATURE VECTOR

We introduce a feature vector $x_t$, reflecting the economic and infrastructure parameters of the digital twin in period $t$:

$$x_t = [P_t, I_t, TR_t, CR_t, EC_t, V_t, \lambda_t, z_t],$$

where: $P_t$ – rental income accrued to the investor; $I_t$ – interest income; $TR_t$ – transaction fees; $CR_t$ – storage fees; $EC_t$ – share of operating expenses; $V_t$ – transfer volume (characteristic of transaction activity); $\lambda_t$ – network congestion parameter; $z_t$ – vector of additional factors (e.g., changes in DFA issuance terms, macroeconomic indicators, volatility parameters).

Then the digital twin of cash flow is defined as a function of the features:

$$\widetilde{CF}_t = g(x_t).$$

## 6.3 NEURAL NETWORK ARCHITECTURE

For forecasting, a multilayer feedforward neural network (in the basic formulation) or a recurrent architecture (in an extended formulation) is used. Within the framework of this model, we fix a universal scheme with input, hidden, and output layers.

Input layer:

$$h_t^{(0)} = x_t.$$

Hidden layers. For layer $l$:

$$h_t^{(l)} = \sigma\big(W^{(l)} h_t^{(l-1)} + b^{(l)}\big),$$

where $W^{(l)}, b^{(l)}$ – weight matrix and bias vector of layer $l$; $\sigma(\cdot)$ – non-linear activation function.

Output layer. Forecast of the digital cash flow:

$$\widehat{\widetilde{CF}}_{t+1} = W^{(L+1)} h_t^{(L)} + b^{(L+1)}.$$

For a forecast over horizon $h$, the forecast is formed iteratively or by a vector output of the network:

$$\widehat{\widetilde{CF}}_{t+1:t+h} = \Psi(h_t^{(L)}).$$

## 6.4 MODEL TRAINING PROCEDURE

Network training is carried out by minimizing the loss function between the actual and forecasted values of the digital cash flow:

$$\mathcal{L}(\theta) = \sum_{t=1}^{T} \big(\widetilde{CF}_t - \widehat{\widetilde{CF}}_t\big)^2,$$

where $\theta$ – the set of network parameters. In the case of high flow volatility, it is possible to use robust loss functions, but in the basic formulation, the quadratic error is used.

## 6.5 SCENARIO-BASED FORECASTING MECHANISM

To account for the uncertainty of the digital environment, a set of scenarios $S$ is formed, differing in key risk factors, primarily in the components $TR_t$, $\lambda_t$, and volatility parameters:

$$\widehat{\widetilde{CF}}_{t+1}^{(s)} = \Phi(x_t^{(s)}), \quad s \in S.$$

Then the scenario with the minimum deviation from the reference trajectory is selected, or an aggregated forecast is calculated:

$$\widehat{\widetilde{CF}}_{t+1} = \sum_{s \in S} w_s \widehat{\widetilde{CF}}_{t+1}^{(s)}, \quad \sum_{s \in S} w_s = 1,$$

where $w_s$ are the scenario weights.

## 6.6 FINAL INTERPRETATION OF THE ALGORITHM

The proposed algorithm for neural network forecasting of the digital twin includes:

1. formation of input features based on transactions and financial parameters;
2. construction of a predictive mapping by a neural network;
3. training the network on historical data of the digital twin;
4. formation of scenario forecasts considering infrastructure risks.

The resulting model ensures the reproduction of the digital twin as a dynamic object of the distributed ledger and allows comparing its characteristics with the classical model of investment cash flow.

# 7 COMPARATIVE ANALYSIS OF THE DIGITAL TWIN AND THE CLASSICAL MODEL OF INVESTMENT CASH FLOW

## 7.1 METHODOLOGICAL APPROACH TO COMPARISON

The comparison of the digital twin of investment cash flow and the classical discounted cash flow model is carried out according to the following criteria:

1. legal structure of the investment;
2. cash flow structure;
3. infrastructure environment of circulation;
4. nature of transaction costs;
5. income generation mechanism;
6. discounting parameters;
7. forecasting algorithm.

This approach allows comparing models not descriptively, but at the level of economic structure and mathematical formalization.

## 7.2 CRITERIA-BASED ANALYSIS

### 7.2.1 LEGAL STRUCTURE

Classical model implies the transfer of ownership of the object to the investor or the emergence of a debt claim.

Digital twin records the investor's rights in the distributed ledger without transferring ownership of the tangible asset. The investment right is realized through a smart contract.

Consequence: the risk structure and distribution of responsibility in the digital model differ from the traditional investment structure.

### 7.2.2 CASH FLOW STRUCTURE

Classical model:
$$CF_t = Inflow_t - Outflow_t.$$

Digital model:
$$\widetilde{CF}_t = P_t + I_t - TR_t - CR_t - EC_t.$$

The difference lies in the appearance of infrastructure fees $TR_t$ and $CR_t$, as well as algorithmically generated income $I_t$.

### 7.2.3 INFRASTRUCTURE ENVIRONMENT

The classical model operates in a centralized accounting system.

The digital twin exists in a decentralized environment, where:

- data storage is distributed among network nodes;
- transaction confirmation is carried out algorithmically;
- settlements are executed by a smart contract.

Consequence: the nature of transaction costs and time lags changes.

### 7.2.4 TRANSACTION COSTS

In the classical model, transaction costs are institutional in nature and associated with the participation of intermediaries.

In the digital model, transaction costs are determined by network parameters and can be expressed by the function:

$$TR_t = f(V_t, \lambda_t).$$

Thus, the fee burden becomes an endogenous parameter of the system.

### 7.2.5 INCOME GENERATION MECHANISM

Classical investment in a productive asset generates operating cash flow.

In the digital twin, the investment flow may include:

- rental income $P_t$;
- interest income $I_t$, calculated automatically.

Consequently, the digital model combines features of investment and financial income.

### 7.2.6 DISCOUNTING PARAMETERS

The classical discount rate is determined based on market models.

In the digital model, the rate

$$r_d = r_f + \pi_m + \pi_{reg} + \pi_{tech}$$

is supplemented by premiums for technological and regulatory risk.

The lack of long-term statistics complicates the estimation of $r_d$.

### 7.2.7 FORECASTING ALGORITHM

The classical model is forecasted using financial analysis methods.

The digital twin is forecasted using a neural network:

$$\widehat{CF}_{t+1} = \Phi(x_t),$$

where the input vector includes infrastructure parameters of the distributed ledger.

Thus, the forecasting algorithm is integrated into the digital environment and possesses adaptive properties.

### 7.3 SUMMARY COMPARATIVE CHARACTERISTICS

Table 1: Comparative characteristics of the classical cash flow model and the digital twin.

| Criterion | Classical Model | Digital Twin |
|---|---|---|
| Ownership | Transfers to investor | Retained by the issuer |
| Fees | Institutional | Infrastructure, network-dependent |
| Income Structure | Operating flow | Operating + algorithmic income |
| Accounting Environment | Centralized | Decentralized |
| Discount Rate | Market models | Supplemented by technological premiums |
| Forecasting | Financial methods | Neural network forecasting |

## 7.4 CONCLUSIONS OF THE COMPARATIVE ANALYSIS

The conducted analysis shows that the digital twin of investment cash flow is not an alternative to the classical model, but rather its structural modification, integrated into a distributed ledger.

The main difference lies in the transformation of:

- the institutional environment of circulation;
- the structure of fee costs;
- the income generation mechanism;
- risk parameters;
- the forecasting algorithm.

Thus, the digital twin forms a new model of investment cash flow, combining elements of financial mathematics, distributed systems, and machine learning.

## 8 PRACTICAL IMPLEMENTATION

Justifying the digital twin model of investment cash flows by decomposing infrastructure fees $TR_t$, assuming the following scenario.

The investor utilizes a blockchain platform where the median transaction fee is 0.001 USD. An empirical median fee, sourced from publicly available data, is used for the calculations. An empirical median fee, for which information is available from public sources, is used for the calculations.

Transaction protection mechanisms do not utilize manual processing batches, with Bitcoin's confirmation time being 400 milliseconds. The maximum block throughput capacity (TPS) is 1000 milliseconds.

No requirement exists for creating a token account. No additional fee is incurred for token account creation.

The rental payment amount for the physical server (64 GB RAM, 3.2 GHz, 6 cores) is equivalent to 75 USD.

The network load parameter $\lambda_t$ is specified by the configuration:

$$\lambda_t = \frac{BT}{TPS}$$

where: BT – transaction confirmation time; TPS – maximum block throughput capacity. By interpolating the values into the expression, we obtain the network load parameter $\lambda_t$ 0.4

The transfer volume (rental payment) takes the form:

$$V_t = \sum_{t=1}^{n} TV - (C - fee)$$

where: n – number of outputs; TV – net rental payment volume; C – unspent remainder from previous transactions; Fee – fee for conducting a single transaction.

Assume that the unspent remainder from previous transactions equals 20 USD, and the number of outputs is 1. Substituting the variable values into the expression yields a transfer volume of 55.001 USD.

The transaction fee in the blockchain, as a variable quantity, takes the form of a function of the transaction volume:

$$TR_t = f(55.001, 0.4).$$

Since the methods for determining the fee function are inherently embedded in the blockchain, we focus on defining the function through the multiplication of variables. In our view, this approach is applicable, as the network load parameter $\lambda_t$ and transfer volume $V_t$ are already defined and require no additional calculations.

We define the transaction volume function as the product of two functions, 55.001 and 0.4. Under the specified criteria of utilizing maximum block throughput capacity during high blockchain network load, we obtain a transaction fee $TR_t$ value of 22.0004 USD.

The performed calculation of transaction fees $TR_t$ yields the following conclusions:

- with significant volumes of concurrent user transactions in the network, the block throughput capacity utilizes maximum production resources, which contributes to an increase in fees;

- a larger volume of unspent remainder from previous transactions reduces the net inflow volume;

- the transfer volume $V_t$ is directly dependent on the network load parameter $\lambda_t$.

We examined a specific case of transaction fees $TR_t$. In the applied context of utilizing a digital cash flow model, various scenarios may arise wherein transaction fees fluctuate—decreasing or increasing—depending on diverse internal and external factors.

## 9   CONCLUSION

In the course of the research, the task of theoretical formalization of the digital twin of investment cash flow operating in a distributed ledger was formulated and solved.

Based on a systematic analysis, it is established that the digital twin is not a simple reproduction of the classical discounted cash flow model, but represents its structurally modified form, integrated into the blockchain infrastructure.

The following main results were obtained in the work.

1. The classical model of investment cash flow is formalized and its basic assumptions are defined: transfer of ownership, centralized accounting, traditional discounting mechanism, and the institutional nature of transaction costs.

2. A mathematical model of the digital twin of investment cash flow is developed, including infrastructure fees of the distributed ledger, algorithmically accrued interest income, decentralized fixation of obligations, and an extended discount rate structure.

3. It is shown that in the digital model, the risk structure undergoes transformation: regulatory and technological components are added to the market premium, which requires modification of approaches to determining the discount rate.

4. An algorithm for neural network forecasting of the digital twin is proposed, based on the formation of a feature vector including both financial and infrastructure parameters of the blockchain system. It is proven that forecasting the digital twin is adaptive in nature and takes into account the dynamics of the transaction environment.

5. A comparative analysis of the classical and digital models is performed, which allowed establishing that the key difference lies not in the discounting formula, but in the institutional and technological transformation of the investment process.

The scientific novelty of the research lies in the development of a formalized model of the digital twin of investment cash flow and its integration with a neural network forecasting algorithm in a distributed ledger environment. For the first time, a structural decomposition of the digital investment cash flow is proposed, highlighting infrastructure and algorithmic components.

The practical significance of the work lies in the possibility of applying the proposed model for the valuation of digital financial assets backed by investments in production facilities of IT infrastructure, as well as for constructing adaptive systems for forecasting their cash flows.

As directions for further research, it is advisable to:

- develop a stochastic version of the digital twin model considering the probabilistic dynamics of network fee parameters;
- formalize a methodology for empirical calibration of the discount rate for digital assets;
- conduct approbation of the neural network algorithm on real data of digital financial assets circulation.

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
