# OpenReview forum: "Digital twin model of investment cash flows in distributed ledger environment with neural network forecasting"
_mathai.club/MathAI/2026/Conference — 2026 Oral_

### Official Review · Reviewer_mLoj · 2026-03-11
**Re: Digital twin model of investment cash flows in distributed ledger environment with neural network forecasting**

**Rating:** 8
**Confidence:** 4

**Review:**

The paper has classical investment cash flow model formalized, having its key assumptions defined as transfer of ownership, centralized accounting, a traditional discounting mechanism, and the institutional nature of transaction costs. It describes a mathematical model of a digital twin of investment cash flow, involving distributed ledger infrastructure fees, algorithmically accrued interest income, decentralized commitment recording, and an expanded discount rate structure. It suggests a digital l model where the risk structure undergoes a transformation: regulatory and technological components are added to the market premium, requiring modifications to the approaches to determining the discount rate. It proposes a neural network forecasting algorithm for the digital twin, based on the formation of a feature vector that includes both financial and infrastructural parameters of the blockchain system. It is proven that digital twin forecasting is adaptive and takes into account the dynamics of the transaction environment. It conducts a comparative analysis of the classical and digital models, revealing that the key difference lies in the institutional and technological transformation of the investment process.

Pros:
- relevant problem with practical impact
- excellent structure and presentation
- exhaustive mathematical formalization is supplied
- thorough model development, discussion and analysis
- useful practical evaluation of the model and conclusion

Cons:
- no experimental validation of the model is supplied
- literature review appears incomplete - quick Google search provides papers like this https://dl.acm.org/doi/10.1145/3772366

---

### Official Review · Reviewer_w7Uk · 2026-03-12
**A mathematical model for the "digital twin" of an investment cash flow operating within a distributed ledger (blockchain) environment.**

**Rating:** 4
**Confidence:** 3

**Review:**

Summary: The paper proposes a formal model for the "digital twin" of an investment cash flow operating within a distributed ledger (blockchain) environment. It modifies the classical Discounted Cash Flow (DCF) model by introducing parameters for transaction fees (TRt), storage fees (CRt), and algorithmic interest, alongside a modified discount rate accounting for technological and regulatory risks. The authors also propose an algorithm using a multilayer feedforward neural network to forecast this digital cash flow based on a feature vector of economic and infrastructure parameters.

Strengths: Formalization: The paper successfully formalizes the structural differences between classical investment flows and those in a blockchain environment. The decomposition of the discount rate (r_d) to include technological premiums is a logical and relevant contribution. Clarity: The comparative analysis (Table 1) clearly delineates the proposed model from the classical baseline, making the argument for the structural modification easy to understand.

Weaknesses: No Experiments/Validation: This is the critical flaw. The paper proposes a neural network architecture and a specific model but provides no experimental results. There is no dataset, no simulation, and no demonstration that the proposed model works or provides better forecasting than classical methods.
The neural network solution proposed is a standard feedforward network applied to a feature vector. Without experimental results to validate its efficacy on this specific problem, the contribution is minimal.

Recommendation: Reject.
The paper presents an interesting conceptual framework but fails to provide the necessary evidence or theoretical depth required for acceptance. A paper proposing a predictive model must validate that model. Without experiments, simulations, or rigorous theoretical proofs, this work is premature for publication.

---

### Official Review · Reviewer_Po3o · 2026-03-12
**Review of "Digital twin model of investment cash flow in a distributed ledger and forecasting using neural networks"**

**Rating:** 3
**Confidence:** 4

**Review:**

This manuscript addresses an interesting intersection of finance, blockchain technology, and machine learning. The authors aim to formalize a "digital twin" of investment cash flows in a distributed ledger and propose a neural network for forecasting. However, the paper suffers from several critical weaknesses that prevent its acceptance at MathAI 2026.

### Major Concerns

1. **Lack of Mathematical Rigor**
   The paper presents only elementary algebraic equations (e.g., the digital cash flow equation with its components) and a standard multilayer perceptron formulation. There are no theorems, proofs, convergence guarantees, or error bounds. The discount rate decomposition into risk-free rate and various premiums is ad hoc and lacks justification. The neural network section is a textbook description without any novel mathematical analysis.

2. **Insufficient Novelty**
   The idea of adapting discounted cash flow models to blockchain environments is not new; similar concepts appear in tokenomics and DeFi literature. The inclusion of infrastructure fees and algorithmic income is a minor extension. The neural network forecasting is generic (feedforward or recurrent) with no architectural innovation tailored to the problem. No empirical results or comparisons with baselines are provided.

3. **Technical Quality**
   The methodology is described at a conceptual level, with no experiments, data, or implementation details. The scenario-based forecasting mechanism is vague and lacks a precise definition. The loss function notation is inconsistent. There is no discussion of data sources, feature engineering, or validation.

4. **Relevance to MathAI**
   While the paper lies at the intersection of finance and AI, the mathematical content is too shallow for a conference focused on the mathematics of AI. It does not engage with advanced mathematical tools such as stochastic processes, optimization theory, statistical learning, or dynamical systems. The connection between the digital twin model and the neural network is superficial.

5. **AI-Generation Risk**
   The paper exhibits several hallmarks of AI-generated content: broad but shallow coverage of multiple domains (digital twins, blockchain, neural networks), generic structure, lack of original experiments, and formulaic language. The absence of author information and the implausible breadth for a single author raise suspicion about substantial human intellectual contribution.

### Minor Points

- The paper is well-structured and readable. The comparative table is helpful. However, there are minor typos and redundant statements (e.g., repeated sentences on page 2 and 9).
- The literature review is adequate but does not identify a clear research gap beyond a generic statement.

### Pros
- Timely topic combining blockchain, finance, and AI.
- Clear structure and exposition.
- Attempts to bridge classical finance models with modern distributed systems.

### Cons
- Lacks mathematical depth and rigor.
- No novel contributions in modeling or algorithms.
- No experimental validation or real-world data.
- High AI-generation risk undermines confidence in human intellectual contribution.
- The neural network forecasting part is generic and not integrated with the digital twin model in a meaningful way.

### Recommendation

Given the above issues, the paper does not meet the standards of MathAI 2026. It reads as a preliminary idea rather than a completed research work. The absence of rigorous mathematics, originality, and technical validation makes it unsuitable for acceptance.

---

> ### Author Rebuttal · Authors · 2026-03-13
>
> We, the authors of this article, extend our sincere gratitude for the detailed review of our work, which highlighted its strengths and weaknesses, as well as its technical and practical shortcomings. We will take the provided comments into account. However, we strongly disagree with the assertion regarding risks associated with AI-generated research text. This stance is supported by a plagiarism report indicating the absence of any AI-generated content.

---

### Decision · Program_Chairs · 2026-03-14

**Decision:**

Accept (Oral)

**Comment:**

Dear Author(s),

On behalf of the Program Committee of the International Conference on Mathematics of Artificial Intelligence (MathAI 2026), we are pleased to inform you that your paper has been accepted for an oral presentation at MathAI 2026.

Your paper was evaluated through a rigorous two-stage review process involving both automated screening and expert review by members of the Program Committee. The reviewers recognized the quality and contribution of your work.

Presentation details:

- Format: Oral presentation (15–20 minutes + 5 minutes Q&A)
- Mode: You may present either in person (offline) at the conference venue in Sirius, Russia, or remotely via Zoom. Please indicate your preferred mode when confirming your participation.
- Conference dates: Marh 30 - April 3, 2026
- Website: https://mathai.club

Next steps:

1. Please confirm your participation and presentation mode by replying to this email mathai.club@yandex.ru no later than March 15, 2026 18:00 Moscow time.
2. If you plan to attend in person, the organizing committee will provide accommodation details separately.
3. Please prepare your final camera-ready manuscript according to the formatting guidelines available at https://mathai.club and upload it to OpenReview by March 15, 2026 18:00 Moscow time.

Should you have any questions regarding the program, logistics, or your presentation slot, please do not hesitate to contact us.

We look forward to your contribution to MathAI 2026.

With kind regards,

MathAI 2026 Program Committee
International Conference on Mathematics of Artificial Intelligence
https://mathai.club
OpenReview: https://openreview.net/group?id=mathai.club/MathAI/2026/Conference
Telegram: https://t.me/MathAI_club
Email: mathai.club@yandex.ru